# Proposed Mechanism for Emodin as Agent for Methicillin Resistant Staphylococcus Aureus: In Vitro Testing and In Silico Study

Mohammed M. Ghoneim [1,2,†], Rasha Hamed Al-Serwi [3,†], Mohamed El-Sherbiny [4], El-sayed M. El-ghaly [1], Amal E. Hamad [5], Mohamed A. Abdelgawad [6], Ehab A. Ragab [1], Sarah I. Bukhari [7], Khulud Bukhari [8], Khaled Elokely [9,*] and Manal A. Nael [9,10,*]

1 Pharmacognosy and Medicinal Plants Department, Faculty of Pharmacy, Al-Azhar University, Cairo 11884, Egypt
2 Department of Pharmacy Practice, College of Pharmacy, AlMaarefa University, Riyadh 11597, Saudi Arabia
3 Department of Basic Dental Sciences, College of Dentistry, Princess Nourah bint Abdulrahman University, Riyadh 11671, Saudi Arabia
4 Department of Basic Medical Sciences, College of Medicine, AlMaarefa University, Riyadh 11597, Saudi Arabia
5 Department of Pharmaceutical Analytical Chemistry, Faculty of Pharmacy, Menoufia University, Shebin El-Koum 32511, Egypt
6 Department of Pharmaceutical Chemistry, College of Pharmacy, Jouf University, Sakaka 72341, Saudi Arabia
7 Department of Pharmaceutics, College of Pharmacy, King Saud University, Riyadh 11451, Saudi Arabia
8 Department of Microbiology and Parasitology, College of Veterinary Medicine, King Faisal University, Hofuf, Al-Ahsa 36362, Saudi Arabia
9 Department of Chemistry, Institute for Computational Molecular Science, Temple University, Philadelphia, PA 19122, USA
10 Department of Pharmaceutical Chemistry, Faculty of Pharmacy, Tanta University, Tanta 31527, Egypt
* Correspondence: kelokely@temple.edu (K.E.); manal.nael@temple.edu (M.A.N.)
† These authors contributed equally to this work.

**Abstract:** In the search for a new anti-MRSA lead compound, emodin was identified as a good lead against methicillin-resistant *Staphylococcus aureus* (MRSA). Emodin serves as a new scaffold to design novel and effective anti-MRSA agents. Because rational drug discovery is limited by the knowledge of the drug target, α-hemolysin of *Staphylococcus aureus* was used in this study because it has an essential role in *Staphylococcus* infections and because emodin shares structural features with compounds that target this enzyme. In order to explore emodin's interactions with α-hemolysin, all possible ligand binding pockets were identified and investigated. Two ligand pockets were detected based on bound ligands and other reports. The third pocket was identified as a cryptic site after molecular dynamics (MD) simulations. MD simulations were conducted for emodin in each pocket to identify the most plausible ligand site and to aid in the design of potent anti-MRSA agents. Binding of emodin to site 1 was most stable (RMSD changes within 1 Å), while in site 2, the binding pose of emodin fluctuated, and it left after 20 ns. In site 3, it was stable during the first 50 ns, and then it started to move out of the binding site. Site 1 is a possible ligand binding pocket, and this study sheds more light on interaction types, binding mode, and key amino acids involved in ligand binding essential for better lead design. Emodin showed an IC$_{50}$ value of 6.3 µg/mL, while 1, 6, and 8 triacetyl emodin showed no activity against MRSA. A molecular modeling study was pursued to better understand effective binding requirements for a lead.

**Keywords:** MRSA; emodin; α-hemolysin; molecular docking; molecular dynamics; cryptic sites

## 1. Introduction

Humanity is facing many challenges that threaten its own existence. At the time of this study, the world was facing one of its worst nightmares—a global pandemic that disrupted

human activities and increased hospitalization and intensive care admission to such a level that it surpassed maximum capacity many times [1–3]. This increased hospitalization brings forward the looming risks of nosocomial infections that could complicate cases, prolong hospital stays and even increase fatalities. Amongst the most notorious nosocomial infections, MRSA (methicillin-resistant Staphylococcus aureus) is a top assailant [4]. Preventive measures have led to a continuous decline in healthcare-related MRSA invasive infections in developed countries; nevertheless, this decline is now slowing down. Moreover, community-based infections, which constitute a reservoir, are not decreasing, and many developing countries still have a stubbornly high prevalence [5]. The high virulence of MRSA and the increased difficulty of treating it leads to a high fatality rate [6]. Another worrying feature is the ease with which it acquires resistance to new antibiotics making its treatment more difficult by the day and increasing its mortality [7].

Amongst its vast and versatile arsenal, cytolytic toxins are especially important for its virulence. α-toxin is an important one that helps MRSA kill leukocytes [8]. This toxin, also referred to as α-hemolysin [9], is a host-specific protein that binds to the cell walls of leukocytes and perforates them, causing cell lysis, thus representing an important therapeutic target in the fight against MRSA. The structure and mechanism of action of α-hemolysin, therefore, have been widely studied [10–12], and it was selected for our molecular modeling study because it is the target of emodin-like compounds [13]. Other anthraquinones have been reported for their anti-MRSA activity [14,15].

Emodin was identified as the active ingredient from Rhizoma Polygoni Cuspidati, which is responsible for significant anti-MRSA activity in the in vitro antibacterial experiments. The study suggested that emodin could damage cell wall integrity [16]. Emodin did not show significant cytotoxicity against human cells. The synthesis and antibacterial evaluations were reported for emodin and several analogs, indicating the importance of the hydroxyl and methyl groups in the emodin scaffold for the anti-MRSA activity [17]. The mechanism of action of emodin is not clear. Proteomic studies were used to explore the molecular mode of action of emodin, showing significant changes in several MRSA proteins, which indicates that emodin could exert its effects by multiple mechanisms [18].

Molecular modeling is used in this work to investigate the binding of emodin and related compounds to three proposed ligand sites in α-hemolysin. Molecular Docking and Molecular Dynamics (MD) simulations were conducted to gain a better understanding of what makes a good fit for α-hemolysin active sites. This is an essential step if we are to successfully develop effective anti α-hemolysin therapeutics that increases activity against MRSA.

## 2. Materials and Methods

*Antimicrobial Assay*

Emodin (Compound **1**) and 1, 6, 8 triacetyl emodin (Compound **2**) were tested for antimicrobial activity. All organisms were obtained from the American Type Culture Collection (Manassas, VA, USA). Fungal strains: *Candida albicans* ATCC 90028, *Cryptococcus neoformans* ATCC 90113, *Aspergillus fumigatus* ATCC 90906, methicillin-resistant *Staphylococcus aureus* ATCC 43300 (MRSA), *Escherichia coli* ATCC 35218, *Pseudomonas aeruginosa* ATCC 27853 and *Mycobacterium intracellulare* ATCC 23068. Ciprofloxacin and amphotericin B were used as positive controls for bacteria and fungi, respectively [19].

The following protocol was followed: The American Type Culture Collection was used to obtain all of the organisms (Manassas, VA, USA). Optical density was employed to track growth for all species, and a modified version of the Clinical and Laboratory Standards Institute (formerly NCCLS) procedures was utilized for susceptibility testing. For growth detection of M. intracellulare and A. fumigatus, 5% Alamar Blue (Biosource International, Camarillo, CA, USA) was added to media (Biosource International, Camarillo, CA, USA). Samples were duplicated and transferred to 96-well flat bottom microplates after being serially diluted in 20% DMSO/saline. By adjusting the OD630 of the cell/spore suspension in the RPMI at pH 4.5 incubation broth for Candida albicans, microbial inocula were created.

For Cryptococcus neoformans, Sabouraud dextrose was used, followed by cation-adjusted Muller–Hinton at a pH 7.3 for MRSA, 5% Alamar Blue (Biosource International, Camarillo, CA, USA) in Middlebrook 7H9 broth with OADC enrichment, pH = 7.3 for M. intracellulare, and 5% Alamar Blue in RPMI at pH 7.3 for A. fumigatus. Each assay contained drug controls, ciprofloxacin (ICN Biomedicals, Solon, OH, USA) for bacteria and amphotericin B (ICN Biomedicals, OH) for fungi. Prior to and following incubation, all organisms were read at 630 nm using a Polarstar Galaxy plate reader (BMG Lab Technologies, Ortenberg, Germany) or 544 ex/590 cm (M. intracellulare, A. fumigatus) using a Biotek Powerwave XS plate reader (Bio-Tek Instruments, Winooski, VT, USA). By taking 5 L from each clear well, putting it to agar, and letting it sit there until growth appeared, it was possible to calculate the minimum fungicidal or bactericidal concentrations. The MFC/MBC is defined as the test concentration at which the organism dies at the lowest level (allows no growth on agar). Samples are first examined in duplicate at a concentration of 50 µg/mL, and percent inhibitions (% inh.) are estimated in relation to the negative and positive controls. Extracts that exhibit inhibition of 50% move on to the secondary assay. Samples dissolved to 20 mg/mL are tested in the Secondary Assay at 200, 40, 8 µg/mL, while samples dissolved to 2 mg/mL are tested at 20, 4, 0.8 g/mL. The Tertiary Assay is used for pure substances with an IC50 of less than 7 µg/mL in the Secondary Assay. The Tertiary Assay involves testing pure chemicals against microorganisms at concentrations of 20, 10, 5.0, . . . , 0.02 µg/mL and calculating IC50s. Using the XLFit fit curve fitting program, all IC50s are determined. The antifungal drug control is amphotericin B, and the antibacterial drug control is ciprofloxacin.

### 3. Molecular Modeling

**Protein Preparation.** The crystal structure of α-hemolysin of *Staphylococcus aureus* (PDB accession code: 6U3T) was obtained from the RSCB protein data bank repository [20]. The Protein Preparation Wizard (PrepWizard) module of Schrodinger suite [21,22] was used to prepare the protein structure. Missing hydrogen atoms were added, and atom bond orders were assigned. Prime was used to fill in missing residues and loops. Protonation and tautomerization states of amino acid residues were assigned for pH 7.4. Water molecules that are not forming at least three hydrogen bonds to non-waters were deleted. Hydrogen bond networks were optimized for pH 7.4. Then, the protein structure was minimized with the OPLS3 forcefield [23].

**Molecular Docking.** Three possible ligand binding sites were used to prepare the receptor grids for docking [24]. Site 1 was defined using bound ligand coordinates. The center of which was determined by the following amino acids: Asn176, Gln177, Trp179, Tyr182, Gln194, Leu195, Met197, Lys198, Thr199 and Agr200. Site 2 was defined according to published reports by the center of the following amino acids: Tyr102, Ar104, Asn105, Ile107, Asp108, Thr109, Glu111, Tyr112, Met113, Ser114, Gly126, Asp127, Asp128, and Ile132. Site 3, in close proximity to site 2, was determined from the molecular dynamics runs, and it was defined as the center of: Ala62, Gly63, Thr60, Gln64, Arg66, Tyr28, Asp29, Lys30, Gly33, Met34, Ala35, Ser221, Ser222 and Gly223. The docking grids were built by Glide [24–27]. LigPrep [28] was used to prepare ligand structures for docking. Standard precision (SP) docking algorithm in Glide was used, and softening of the receptor potential was allowed to add more receptor flexibility during the docking step. The per-atom van der Waals radii and charges were scaled to 0.85. The five best poses were kept for further calculations. Prime [29–31] was used to refine the docking poses, and more receptor flexibility was allowed for residues within 10 Å of docked ligand. Prime MM-GBSA was then used to calculate the ligand binding free energy (ΔG). Another run of induced fit docking (IFD) was performed using the same amino acids as the center for each site [32–35].

**Molecular Dynamics Simulations.** MD simulations were carried out in triplicates. Three MD systems were built (Table 1). Selected docking poses were solvated in the TIP4P water box [36] and then neutralized with Na+ ions. DESMOND [37] with the OPLS3 force field was used to run the simulations. NPT ensemble at a constant pressure of 1 bar and

temperature of 300 K was set for the calculations by using the Nosé–Hoover chain and Martyna–Tobias–Klein coupling schemes [38–40]. Long-range electrostatic energy and forces were calculated using particle-mesh-based Ewald method [41], and a tolerance was set to $1e^{-9}$. A variant of the SHAKE algorithm was used to allow the time step to be increased. A time-scale splitting (RESPA-based) integrator was utilized for the numerical integration [37]. The short range and bonded interaction were updated every 2ps. The long-range and non-bonded interactions were revised every 6ps. A cutoff of 9.0 Å was defined for the short-range Coulomb interactions.

**Table 1.** System details for MD simulation.

| Binding Site | Simulation Time (ns) | No of Atoms | No of Water Atoms | Counter Ion Conc. mM |
|---|---|---|---|---|
| Site 1 | 100 | 46,683 | 14,022 | 3.890 |
| Site 2 | 100 | 35,658 | 10,347 | 5.272 |
| Site 3 | 100 | 40,926 | 12,103 | 4.507 |

## 4. Results

α-Hemolysin was identified as a target for MRSA because it plays an essential role in *S. aureus* infections. The X-ray crystal structure of the α-hemolysin protein was resolved as a monomer, which assembles to form a heptameric pore (Figure 1). The assembly is a mushroom-shaped structure. The secondary structure of each monomer is made up of twelve β-sheets and one α-helix.

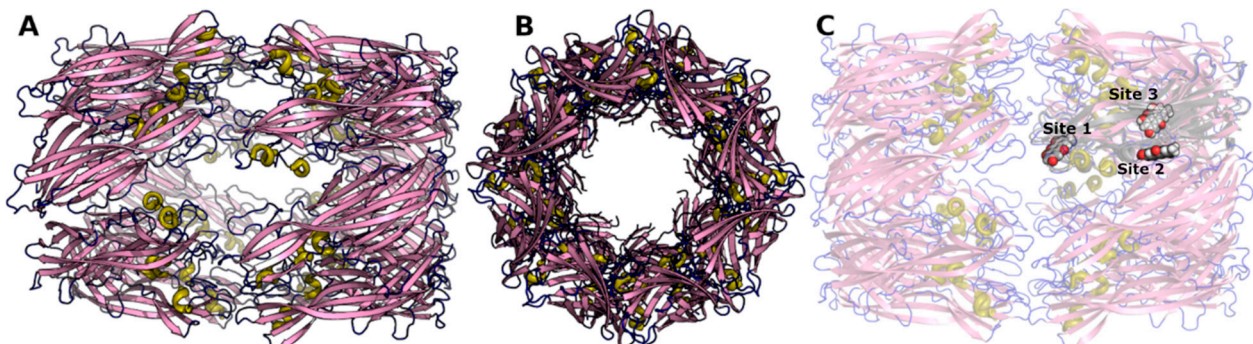

**Figure 1.** α-Hemolysin structure. (**A**). A side view. (**B**). A top view. (**C**). The possible ligand binding sites occupied by emodin in a single monomer. Protein is shown as a cartoon and ligand as spheres.

Compound **1** (Figure 2): emodin (Sigma Aldrich) was evaluated for its antimicrobial activity against MRSA, and it exhibited an inhibitory activity with an IC$_{50}$ value of 6.3 μg/mL (Table 2). Emodin was subjected to acetylation using pyridine and acetic anhydride to give 1, 6, 8 triacetyl emodin (compound **2**), which led to loss of the activity against MRSA.

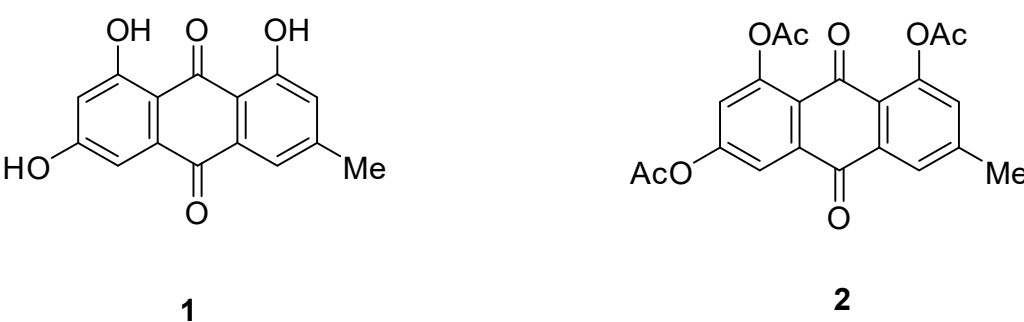

**Figure 2.** Structure of Compounds **1** and **2**.

**Table 2.** Antibacterial activity of compounds **1** and **2**.

|  | Compound 1 IC$_{50}$ (μg/mL) | Compound 2 IC$_{50}$ (μg/mL) | Ciprofloxacin * IC$_{50}$ (μg/mL) |
|---|---|---|---|
| *Staphylococcus aureus* | 4.2 | NA | 0.1 |
| methicillin-resistant *S. aureus* | 6.3 | NA | 0.1 |
| Test concentrations (μg/mL) | 20, 4, 0.8 | 20, 4, 0.8 | 20, 4, 0.8 |

* Ciprofloxacin was used as positive control; NA = not active up to the maximum dose tested 20 μg/mL. Compounds **1** and **2** revealed no activity against *Candida albicans, Cryptococcus neoformans, Aspergillus fumigatus, Escherichia coli, Pseudomonas aeruginosa,* and *Mycobacterium intracellulare*.

To explore a good understanding of the role of hydroxyl groups; a computational study has been performed for both emodin and 1, 6, 8 triacetyl emodin.

The ligand binding pockets in α-hemolysin were explored and defined based on the atomic data of protein complexed with ligands and previously studied sites. Site 1 was determined using ligand coordinates and the center of amino acids that surround the ligand. Site 2 was determined from previous reports [20]. Emodin docked with high affinity into site 1 and site 2. Because emodin is a rigid structure and the identified binding cavities are not wide, an induced fit docking protocol was attempted to provide an insight into the most reliable binding mode in each site. In the case of site 1 (Figure 3), emodin was docked with an IFD score of −661.05 kcal/mol. It exhibited hydrogen bonds with Gln194, Met197, and Arg200. It showed π–π stacking with Trp179, and hydrophobic contacts with several amino acids. Eomdin was docked with an IFD score of −660.45 kcal/mol at site 2. It displayed hydrogen bonds with Asn105, Met113, Ser114 and Asp127, and hydrophobic contacts with several other amino acids. Emodin, in site 3, docked with an IFD score of −524.37 kcal/mol, and it exhibited hydrogen bonds with Gln64, Ser221 and Ser222.

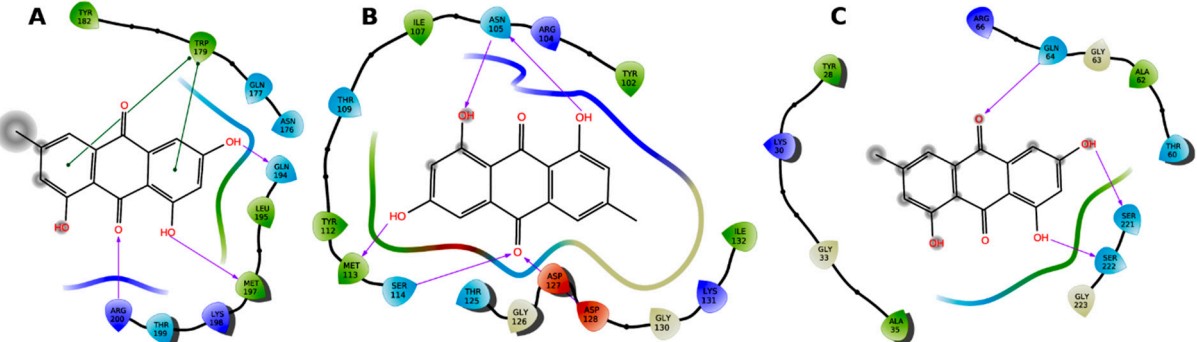

**Figure 3.** The interaction of emodin with the amino acids in site 1 (**A**), site 2 (**B**) and site 3 (**C**). Hydrogen bonds are shown as arrows and π–π stacking as green lines.

Three systems of molecular dynamics (MD) simulations were constructed. The first MD system was built for the best binding mode of emodin in site 1, and the second system was for site 2. Molecular dynamics (MD) simulations were carried out for 100 ns to study the stability of emodin's binding poses. The root mean square deviations (RMSD) were computed according to the following equation.

$$RMSD_x = \sqrt{\frac{1}{N}\sum_{i=1}^{N}\left(r'_i(t_x) - r_i(t_{ref})\right)^2}$$

$N$: the system's number of atoms, $t_{ref}$: the reference time, $r'_i$: the position of protein backbone and ligand atoms (*i*) in frame *x*, $t_x$: the record time of frame *x*.

The average change in displacement of emodin's atoms throughout the MD simulation in site 1 with respect to the original starting point indicated a stable pose in the binding pocket over the course of the MD time. It showed an RMSD value of ~2.0 Å compared to the 1.5 Å of the protein peptide chain (Figure 4A). While in site 2, the binding pose

of emodin fluctuated with an average RMSD value of ~40.0 Å with respect to that of the protein backbone of 2.4 Å (Figure 4B). Emodin left site 2 after 20 ns. Analysis of the MD simulations showed that the loop, which is composed of amino acids 118 through 143, fluctuated the most during the simulation. Emodin left site 2 and occupied another cryptic cavity that may accommodate ligands. Therefore, emodin was docked into site 3 and the best pose was used to construct the third MD system. Emodin fluctuated throughout the MD simulations with an average RMSD of 3.0 Å with respect to the protein backbone of 2.0 Å. It was stable during the first 50 ns, and then it started to move out of the binding site.

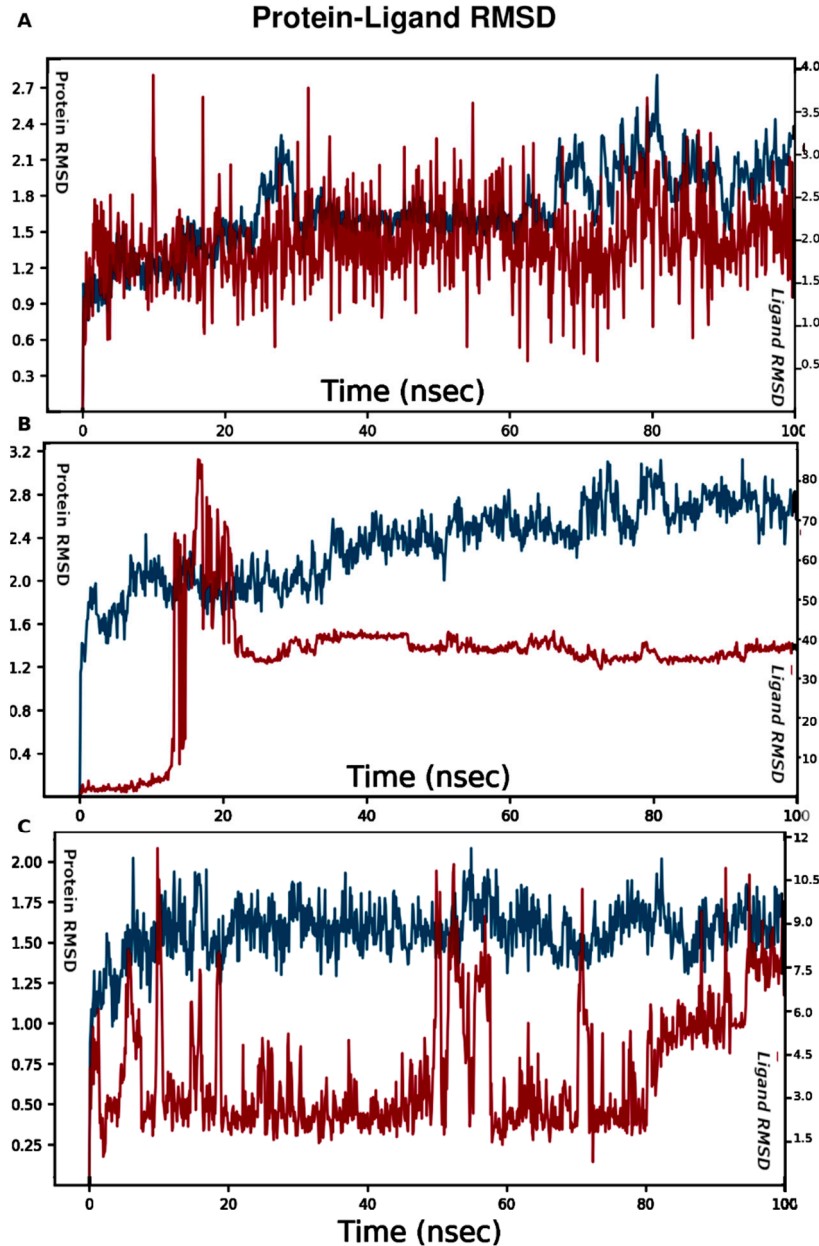

**Figure 4.** RMSD of emodin pose in site 1 (**A**), site 2 (**B**), and site 3 (**C**). The left Y axis is for protein RMSD, and the right scale is for ligand RMSD. How stable a ligand is in relation to a protein and its binding pocket is shown by the ligand RMSD (right Y-axis). Red for the protein backbone and blue for the ligand RMSD. MD simulations time in ns (X-axis).

Emodin is a planar molecule that contains several polar atoms. Interactions of emodin with protein were monitored throughout the simulation. Emodin interacts with the amino acids in site 1 through hydrogen bonds, π–π contact and water-mediated interactions

(Figure 5). On average, emodin has three contacts (Figure 6). The most stable contacts were indicated with Met 197 and Arg200, followed by Asn178, Trp179, and Gly180 (Figure 6).

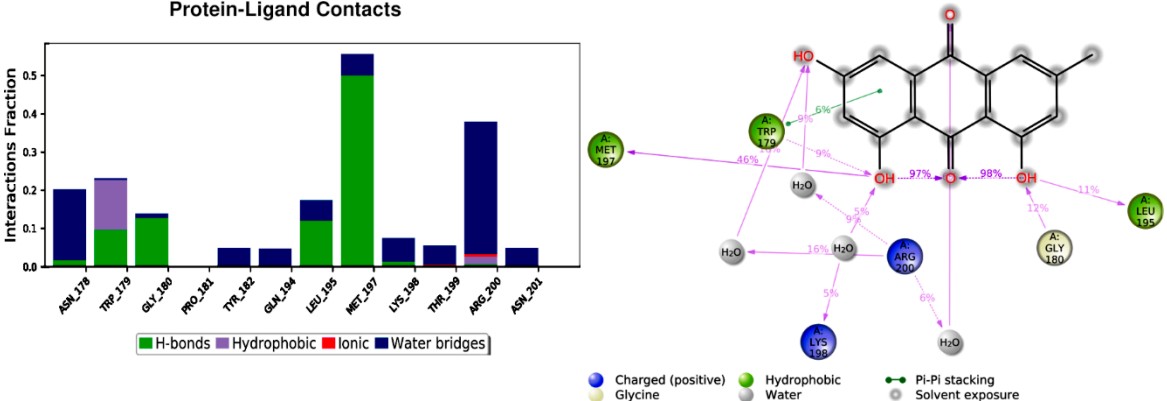

**Figure 5.** Emodin contacts with amino acids in site 1.

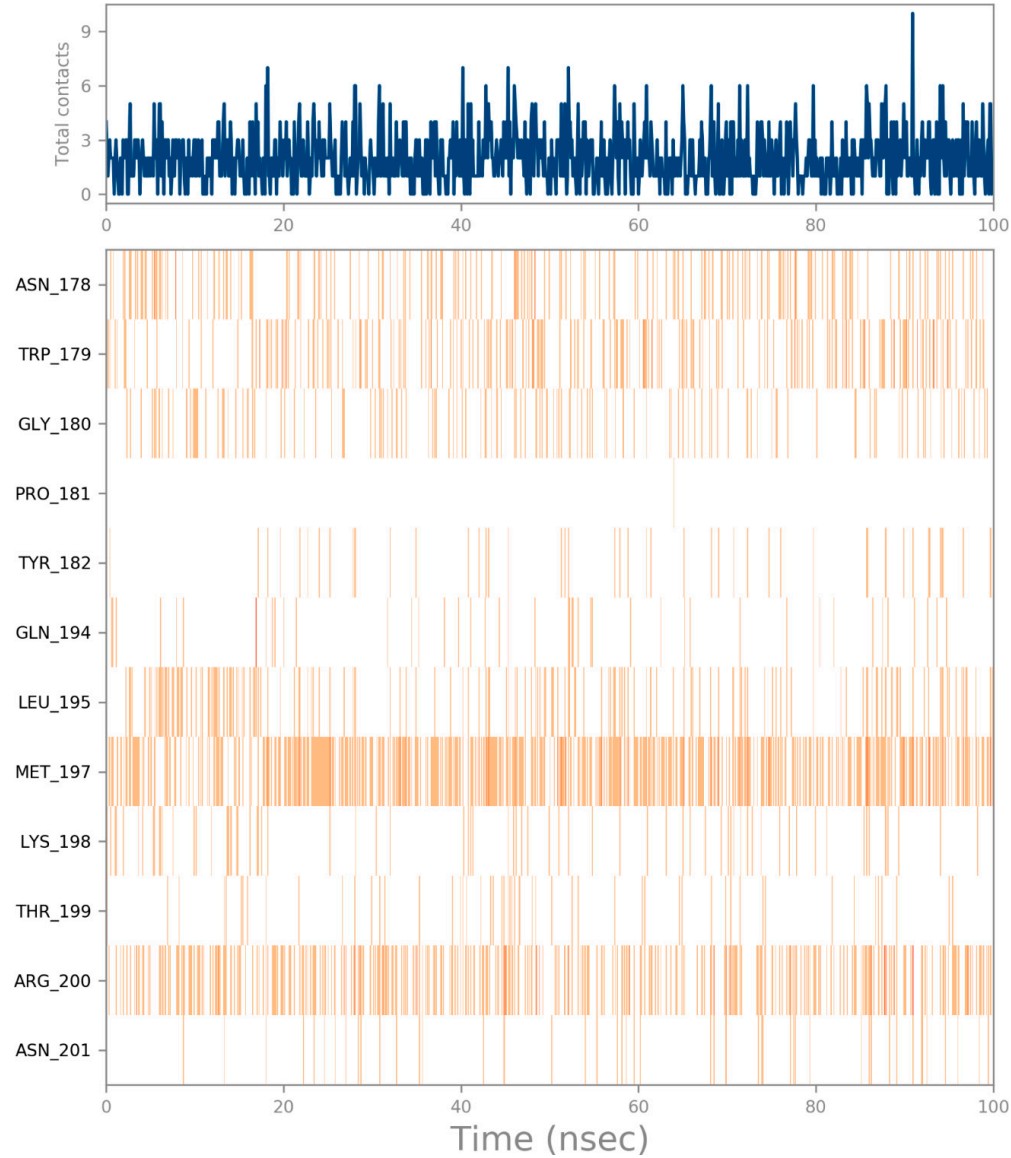

**Figure 6.** Average contacts of emodin in site 1.

In site 2, emodin did not show stable contact with the amino acids in this site. Instead, it displayed interactions with amino acids in site 2 and site 3 as well (Figure S7). During the first 20 ns, only Met113 showed a stable contact, which was lost during the second 80 ns. Emodin started to form more contacts with Tyr65, Val67, Asp212, and Pro213 during the last 80 ns (Figure S8). Emodin was more stable in site 3 in comparison to site 2 (Figure S9). Ser222, Ala62, Gly63, and Tyr28 demonstrated the most stable interactions (Figure S10).

Compound 2 did not show any good binding mode in either site. This could be attributed to extra molecular volume added by the acetyl groups and the block of the interactions brought by the hydroxyl groups of emodin. The planar features of emodin do not make the compound a perfect lead for modification. Understanding the interaction types, binding mode, and recognizing the key amino acids involved in ligand binding would lead to a better lead design. Modifying the scaffold and keeping the important functional groups of emodin would increase ligand affinity and would provide good candidates to manage resistant *Staphylococcus aureus*.

## 5. Conclusions

In this study, emodin was identified as a potential antimicrobial agent against MRSA (IC$_{50}$ of 6.3 μg/mL). However, being a planar compound, it is not a good candidate for a lead modification. Instead, studying the differences between its binding to α-hemolysin and that of the inactive 1, 6, 8 triacetyl emodin serve to better understand binding modes, sites and interaction types. This is an important step in the quest to develop successful anti-MRSA leads. Three possible binding sites were explored for α-hemolysin using MD simulations. Only site 1 is expected to be the possible ligand binding pocket. Finding out the ligand pocket is essential in the stage of potent lead design and modification to develop potent anti-MRSA agents.

**Supplementary Materials:** The following are available online at https://www.mdpi.com/article/10.3390/cimb44100307/s1, NMR data and MS data of compounds (**1** and **2**) are included as Supplementary Material.

**Author Contributions:** Conceptualization, M.M.G., K.E. and M.A.N.; methodology, All authors; software, K.E. and M.A.N.; validation, All authors; formal analysis, All authors; investigation, All authors; resources, All authors; data curation, All authors; writing—original draft preparation, M.M.G., A.E.H., K.E. and M.A.N.; writing—review and editing, All authors; visualization, All authors. All authors have read and agreed to the published version of the manuscript.

**Funding:** The work was supported by Princess Nourah bint Abdulrahman University Researchers Supporting Project number (PNURSP2022R199), Princess Nourah bint Abdulrahman University, Riyadh, Saudi Arabia. This publication was supported by AlMaarefa University researchers supporting program (grant number: MA-006), AlMaarefa University, Riyadh, Saudi Arabia. This project includes calculations carried out on HPC resources supported in part by the National Science Foundation through major research instrumentation grant number 1625061 and by the US Army Research Laboratory under contract number W911NF-16-2-0189.

**Institutional Review Board Statement:** Not applicable.

**Informed Consent Statement:** Not applicable.

**Data Availability Statement:** Not applicable.

**Conflicts of Interest:** The authors declare no conflict of interest.

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
