# Peer review of "Proposed Mechanism for Emodin as Agent for Methicillin Resistant Staphylococcus Aureus: In Vitro Testing and In Silico Study"

_cimb, doi:10.3390/cimb44100307_

Round 1

Reviewer 1 Report

Review comments for the manuscript cimb-1926674

line 91

What are the compounds 1 and 2 ? They are not shown at this point of the manuscript, which is not kind for readers.

In Materials and Methods, Antimicrobial assay

Describe experimental protocols for antimicrobial tests.

Table 2

Only antibacterial test results for S. aureus and MRSA were shown in the Table 2. What about the other bacteria and fungi?

Figs. 3 and 5

The figures are in pale colors difficult to grasp. Please modify them.

Fig. 4

No label and dimension for Y axis. In the figure caption, what is "the right X axis" ?

Recommendation: minor revision required

Author Response

Response

We greatly appreciate the constructive comments and suggestions provided by reviewers, and we have revised our manuscript to address all comments and concerns on a point-by-point basis and are presented in the following pages.

Reviewer # 1

Comment: line 91

What are the compounds 1 and 2 ? They are not shown at this point of the manuscript, which is not kind for readers.

Response: We thank the reviewer for his comments. The comments were addressed and we changed them to Emodin and 1, 6, 8 triacetyl emodin

Comment: In Materials and Methods, Antimicrobial assay

Describe experimental protocols for antimicrobial tests.

Response: The antimicrobial protocol has been added.

Comment: Table 2

Only antibacterial test results for S. aureus and MRSA were shown in the Table 2. What about the other bacteria and fungi?

Response: Thank you for this comment and required data has been added under the table. “Compounds 1 and 2 revealed no activity against Candida albicans, Cryptococcus neoformans, Aspergillus fumigatus, Escherichia coli, Pseudomonas aeruginosa and Mycobacterium intracellulare”

Comment: Figs. 3 and 5

The figures are in pale colors difficult to grasp. Please modify them.

Response: We added modified figures with better contrast.

Comment: Fig. 4

No label and dimension for Y axis. In the figure caption, what is "the right X axis" ?

Response: The labels, dimensions, and axis description are adjusted.

Reviewer 2 Report

It seems that (upon request of the editor?), the authors have chosen to submit their manuscript with a different journal (cibm instead of Int. J. Mol. Sci).

The work gives some important insight in the mechanism of action of emodin, a potential antimicrobial agent against MRSA.

Abstract

Several times, staphylococcus to be changed to Staphylococcus

Line 43, most stable, can this be quantified?

Line 47, 6.3 mg/mL, while

Line 48, A study …

Materials and Methods

Line 95, Coli

Line 97, obtained from?

Line 100, Staphylococcus

Results

Line 144, two times twelve?

Line 153, mL

Line 205, site

Line 215, manage resistant

References

Some are not yet in the style prescribed by the journal: e.g. ; after each author; abbreviated journal name s to be used, only journal volumes to be indicated but not issues

Supplementary materials

Figures S2 and S3, the figure captions are there, but the spectra are missing

Author Response

Response

We greatly appreciate the constructive comments and suggestions provided by reviewers, and we have revised our manuscript to address all comments and concerns on a point-by-point basis and are presented in the following pages.

Reviewer # 2

Comment: Abstract

Several times, staphylococcus to be changed to Staphylococcus

Response: We thank the reviewer for his comments. Staphylococcus has been corrected throughout the text.

Comment: Line 43, most stable, can this be quantified?

Response: We added the RMSD fluctuation value as a measure of stability.

Comment: Line 47, 6.3 mg/mL, while

Response: Corrected.

Comment: Line 48, A study

Response: Corrected.

Comment: Materials and Methods

Line 95, Coli

Response: We thank the reviewer for his comments. E. coli, the species written by convention in lowercase.

Comment: Line 97, obtained from?

Response: We added the antimicrobial protocol with the details of the sources.

Comment: Line 97, obtained from?

Response: We added the antimicrobial protocol with the details of the sources.

Comment: Line 100, Staphylococcus

Response: Corrected.

Comment: Results

Line 144, two times twelve?

Response: “Two” has been deleted.

Comment: Line 153, mL

Response: Corrected.

Comment: Line 205, site

Response: Corrected.

Comment: Line 215, manage resistant

Response: Corrected.

Comment: References

Some are not yet in the style prescribed by the journal: e.g.; after each author; abbreviated journal name s to be used, only journal volumes to be indicated but not issues

Response: References had been corrected

Comment: Supplementary materials

Figures S2 and S3, the figure captions are there, but the spectra are missing

Response: Spectra had been added

Round 2

Reviewer 1 Report

Review comments for the manuscript cimb-1926674

The manuscript has been finely revised and has enough quality to be published.

Recommendation: accept